# Maternal depression in Latinas and child socioemotional development: A systematic review

**Rebeca Alvarado Harris**[1], **Hudson P. Santos, Jr**[1,2]*

1 School of Nursing, University of North Carolina at Chapel Hill, Chapel Hill, North Carolina, United States of America, 2 Institute for Environmental Health Solutions, University of North Carolina at Chapel Hill, Chapel Hill, North Carolina, United States of America

* hsantos@unc.edu

**Data Availability Statement:** All data is presented within the manuscript

**Funding:** The author received no specific funding for this work. HS academic effort was funded by

## Abstract

### Background

Although substantial research exists on the debilitating effects of maternal depression on child development, little is known about Latina mothers with depression and their young children within the broader context of sociocultural and economic stressors.

### Objectives

What is the relationship between maternal depression in Latina mothers and their children's socioemotional outcomes through early developmental windows (0–5 years)?

### Methods

We searched electronic databases PubMed, CINAHL, and PsycINFO in this systematic review, pre-registered via PROSPERO (CRD42019128686). Based on pre-determined criteria, we identified 56 studies and included 15 in the final sample. After extracting data, we assessed study quality with the National Heart, Lung, and Blood Institute Quality Assessment Tool for Observational Cohort and Cross-Sectional Studies.

### Results

We found inverse correlations between maternal depression and child socioemotional outcomes; furthermore, we found evidence of a moderating and mediating role of maternal depression between contextual stressors and child outcomes. Children of U.S.-born Latina mothers had poorer developmental outcomes than children of foreign-born Latina mothers across socioemotional domains and throughout early developmental windows.

### Conclusions

Future research must examine underlying mechanisms for the potential Latino paradox in young Latino children's socioemotional outcomes. Policies should support mental health of Latina mothers as early as the prenatal period.

National Institute of Nursing Research (1K23NR017898-01). The funders had no role in study design, data collection and analysis, decision to publish, or preparation of the manuscript.

**Competing interests:** The authors have declared that no competing interests exist.

## Introduction

Substantial research exists on maternal depression and childhood poverty and their debilitating effects on both maternal health and early child development. Although most of the research implicating the pathways by which maternal depression disrupts child development has been conducted on European-White descendent families [1], evidence suggests that the mother-child relationship is situated within the economic *and* sociocultural contexts in which families live [2]. As such, the intersection between maternal depression and poverty may affect children from ethnic minorities differently. Although Latino families are affected disproportionately by poverty [3], we found few research studies and no systematic reviews on the effect of maternal depression on child development within this vulnerable population. Latina mothers living in poverty experience numerous stressors related to both social adversity and ethnic minority status and are subsequently susceptible to depressive symptomology [4]. Prevalence rates of depression (mild to severe) in Latina mothers range from 12 to 59% in the perinatal period, compared to 10–15% in the general population [5]. Moreover, lack of accessible, affordable, and culturally competent mental health care for Latina mothers has been well-documented [6]. Little is known, however, about the extent to which mental health in Latinas is related to their children's socioemotional development. This is the first systematic review to consolidate studies examining the relationship between Latina mothers' depression and child socioemotional outcomes. Based on a social-ecological perspective, our review aims to study interpersonal and contextual pathways linking maternal depression in Latinas to a young child's (birth to five years old) socioemotional outcomes in the broader context of sociocultural and economic stressors (**Fig 1**).

### Socioemotional outcomes

Socioemotional competencies underlie early childhood mental health and encompass "a child's developing capacity from birth through five years of age to form close and secure adult and peer relationships; experience, regulate, and express emotions in socially and culturally appropriate ways; and explore the environment and learn—all in the context of family, community, and culture." [7] (p. 2). The relationship between maternal depression and disruptions in a young child's socioemotional development and related behavioral manifestations has been well replicated throughout infancy and early childhood [1,8].

Our review focuses on adverse early social and emotional outcomes associated with maternal depression and greater risk for psychopathology later in life [9]: 1) temperamental traits (e.g., higher negative emotionality and lower self-regulatory capacity) in infants that increase vulnerability to stressful caregiving environments; 2) infant social withdrawal behavior; 3) insecure mother-child attachment [10]; and 4) internalizing (e.g., fearfulness, anxiety) and externalizing (e.g., aggression) behaviors in toddlers and preschoolers [9]. We focus on maternal rather than paternal depression due to the high prevalence of maternal depression during sensitive developmental periods [11] and fetal exposure to prenatal depression during pregnancy.

### Maternal sensitivity

Prenatally, maternal depression may alter biochemical pathways associated with physiological and emotional regulation, thus, undermining the foundation on which children develop socioemotional competencies [12,13]. Adverse outcomes, however, may still be mitigated by nurturing mother-child interactions [14]. The early social and cultural caregiving environment continues to shape a young child's socioemotional developmental trajectory and, thereby, promotes future resiliency or exacerbates risk for mental health disorders over the life course [15].

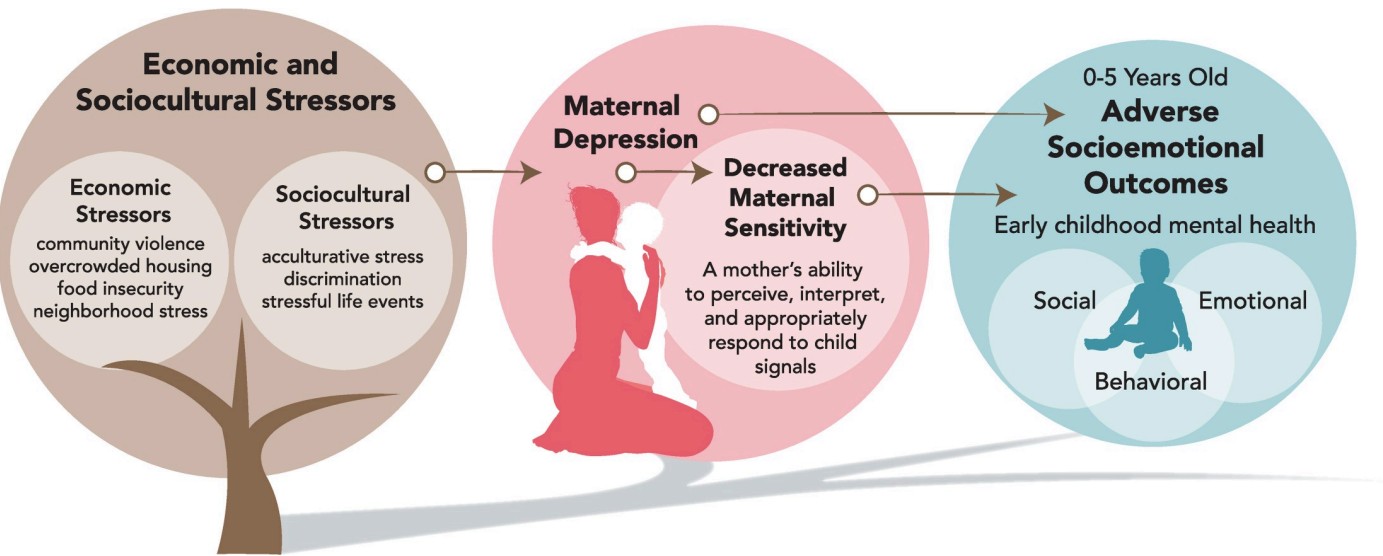

**Fig 1. Potential pathways linking maternal depression in Latinas to child socioemotional outcomes.**

The influence of maternal depression on parenting behaviors (e.g. maternal sensitivity) may underlie the relationship between maternal depression and child socioemotional maladjustment [16,17]. Mothers who are depressed may be less able to provide sensitive interactions needed to develop and sustain early socioemotional competencies, *especially* in the context of economic stressors [18,19].

Given that cultural beliefs and values influence parenting and subsequent child development [20,21], the interaction between maternal sensitivity and maternal mental health on young children's socioemotional outcomes may differ based on sociocultural context [2,22,23]. In fact, similar parenting behaviors often result in contradictory socioemotional child outcomes depending on the cultural meaning underlying the behavior [20]. The effects of maternal depression on maternal sensitivity and subsequent mother-child interactions are both a *modifiable* and *necessary* target for intervention [24]. Empirical evidence suggests that integrated interventions during the first five years of life to reduce maternal depression and enhance parenting skills may ultimately improve children's socioemotional developmental trajectories [16], particularly for the most vulnerable mother-child dyads [25]. However, culturally relevant interventions necessitate further research into the intersection of these variables and related child outcomes within the broader sociocultural and economic context.

## Economic and sociocultural stressors

The relationship between mother and child occurs within the context of poverty in >26% of Latino families with young children [26]. Poverty-related stressors, which have also been used as a proxy for poverty, encompass numerous risk factors for children: food insecurity, overcrowded housing, and community violence. In fact, meta-analyses within the general population have shown that poverty heightens the significant positive association between maternal depression and children's socioemotional difficulties [1,27]. Ethnic minority status also magnifies these effects, but understudied social and cultural pathways underlying such disparities have not been disentangled [1].

Theoretical frameworks such as the Family Stress Model (FSM) hypothesize that economic stressors engender parental stress and depression, compromising parental quality which could

otherwise serve as a child's protective buffer from poverty; as such, maternal psychological functioning may serve as a mediating mechanism by which poverty-related stressors predict child maladjustment [28]. The few FSM studies examining Latino children have focused primarily on middle childhood and adolescence and provide evidence for the mediating role of maternal depression between poverty-related stressors and socioemotional child outcomes [29,30] as well as revealing culture-specific ways in which these variables may interact [31,32]. However, not only do we know little about the validity of these models for Latino children during early childhood [33] but culture-specific stressors have rarely been used as anchors for these studies–despite the integral role of cultural influences on developmental processes [34,35].

Culture-specific stressors (e.g., discrimination, stressful life events, and acculturative stress) have predicted parental depression [36,37] and provide key sociocultural contexts in which to examine maternal and child health [38]. Acculturative stress, stemming from psychosocial adjustment to a host culture, may affect both immigrants and later generations who have simultaneously assimilated to a new culture and maintained cultural heritage [39]. In immigrant women, acculturative stress is often accompanied by family separation, social and linguistic isolation, and discrimination [40]—salient contributors of perinatal depression among Mexican-American mothers [41]. The cumulative effects of these stressors on the relationship between maternal mental health and the child's gestational and early childhood experiences may alter not only the child's life course but the physiology of subsequent generations via biological (e.g., epigenetic) mechanisms [38,42,43] and interpersonal (e.g., parenting) processes [42]. The Latino paradox, wherein less acculturated Latinos experience better health outcomes than more acculturated Latinos, has been documented for maternal and child health: birth weights [44] and breastfeeding rates [45]. However, investigators have not, to our knowledge, explored the relationship between maternal biopsychosocial functioning (e.g., depression) and early child socioemotional outcomes in the context of the Latino paradox.

### Current study

Although the Latino population comprises the largest ethnic minority group in the US [3], young Latino children remain an understudied population. For public policies to be effective, we need to know what harms and what protects children most profoundly amid the economic and sociocultural contexts in which their lives unfold. Research relevant to understanding Latina mothers and their young children may inform interventions that meaningfully support mothers struggling with depression, thereby potentially minimizing or even averting a developmental cascade that undermines critical socioemotional competencies formed early in life. Therefore, this systematic review addresses the following question: during the foundational years from the prenatal period to early childhood (0–5 years), how is maternal depression in Latina mothers related to their children's socioemotional outcomes?

### Methods

This systematic review is pre-registered via PROSPERO (CRD42019128686) and follows the guidelines recommended in the Preferred Reporting Items for Systematic Reviews and Meta-Analyses (PRISMA) [46]. Ethical approval was not required because data was synthesized from previously published studies.

### Study inclusion criteria

We screened studies for eligibility based on predetermined inclusion criteria per the study protocol. To be considered for inclusion, the study needed to assess the relationship between

maternal depressive symptoms and children's socioemotional outcomes within Latina mother-child dyads living in the US. We used the term "Latina" to refer to a female of Latin American origin or descent (e.g., Mexico, Cuba, Puerto Rico, South and Central America), regardless of race. Maternal depression could encompass depressive symptoms on a continuum, elevated scores on self-report measures, or clinical diagnosis of a depressive disorder. We did not exclude for co-morbid symptoms or diagnoses such as anxiety, stress, or another form of psychological distress. Self-report depression measures (e.g. CES-D) often used in population-based studies may more broadly capture psychological distress and diverse affective symptoms (e.g. anxiety) rather than symptoms specific to a depressive disorder [18,47,48]. We defined socioemotional development as a child's emerging capacity for the emotional, behavioral, and social competencies underlying early childhood mental health [9]. Further inclusion criteria were a) young children exposed to maternal depression at any point prenatally up until age five, allowing us to focus on foundational sensitive early developmental periods; b) adult mothers ≥ 18 years old; and c) specified maternal economic status. Studies of subpopulations with confounding secondary diseases or conditions altering maternal depressive symptoms or socioemotional outcomes (e.g., cancer, natural disasters, or developmental delays) were excluded. There were no time restrictions applied to ensure comprehensive results; we completed the search in October 2018. Studies could be in English or Spanish.

## Search strategy

We identified studies through manual searches and through PubMed, CINAHL, and PsycINFO. We applied a comprehensive search string to all three databases to capture studies examining the impact of maternal depression on young children's socioemotional and cognitive outcomes in the Latino population: (Postpartum OR Depression OR Depressive Disorder OR Depress*) AND (Postpartum Period OR Postnatal Care OR Prepartum OR pre-partum OR prenatal OR pre-natal OR antenatal OR perinatal OR peripartum OR postpartum OR post-partum OR postnatal OR post-natal OR puerperium OR parturition OR pregnanc* OR pregnant or mother* OR newborn*) AND (Hispanic Americans OR Latina* OR Latino* OR Hispanic*) AND (child* OR infant OR toddler OR preschooler OR pediatric) AND (develop* OR behavior* OR cognitive OR emotion* OR conduct OR language OR temperament OR socio* OR regulation OR internalizing OR externalizing OR psych*). We manually searched reference lists from relevant literature for any additional eligible studies.

## Study selection and data extraction

We selected studies using the four-phase process for systematic reviews recommended by the PRISMA Group [46] (**Fig 2**). From the electronic database search, we identified 914 studies after removing 466 duplicates. We screened titles and abstracts from identified studies for eligibility and excluded 858 studies not meeting inclusion criteria. We retrieved the full text of the 56 remaining to determine eligibility, and recorded reasons for exclusion for the 38 non-eligible studies. In the final phase, 17 studies from the database search and one study from the manual search met all eligibility criteria and were selected for data extraction. Data from included literature was extracted onto a template which included study purpose, first author, publication year, study design, sample characteristics, maternal depression measures and data collection points, socioemotional or cognitive outcome measures and data collection points, main analytical approach and covariates, and main results. When available, country of origin and acculturation data was extracted. A second reviewer evaluated the extracted data for thoroughness. During the extraction phase, two studies did not include maternal age ranges, and three studies included broad maternal age ranges that extended into adolescence. Reviewers



# PRISMA 2009 Flow Diagram

**Identification**

Records identified through
database searching
(n = 1380)

Additional records identified
through other sources
(n = 1)

Records after duplicates removed
(n = 914)

**Screening**

Records screened
(n = 914)

Records excluded
(n = 858)

**Eligibility**

Full-text articles
assessed for eligibility
(n =56)

Full-text articles
excluded, with reasons
(n =41)
16 Child socio-emotional
and/or cognitive outcome not
available
10 Does not differentiate
between ethnicity/race
outcomes
6 Does not include maternal
depression as a distinct risk
factor
3 Depression associations
not stratified
3 Removed cognitive
outcomes from final results
1 Does not specify economic
status
1 study in middle childhood
1 full text not found

**Included**

Studies included in
qualitative synthesis
(n = 15)

*From:* Moher D, Liberati A, Tetzlaff J, Altman DG, The PRISMA Group (2009). *Preferred Reporting Items for Systematic Reviews and Meta-Analyses: The PRISMA Statement. PLoS Med 6(7): e1000097. doi:10.1371/journal.pmed1000097*

**For more information, visit www.prisma-statement.org.**

**Fig 2. PRISMA flow chart diagram.**

agreed to include these studies; the broad age range would provide a representative sample of Latina mothers in the United States. We contacted authors for maternal age distribution if needed. Moreover, we included three dissertations meeting inclusion criteria.

We extracted data for cognitive outcomes because they are interwoven with developing socioemotional competencies early in life [9]. However, these were excluded from our final qualitative synthesis and placed in a supplemental table in order to focus our attention on early social and emotional outcomes (S1 Table).

### Quality assessment

Study quality (**Table 1**) was assessed by two reviewers following the guidelines established in the National Heart, Lung, and Blood Institute (NHLBI) Quality Assessment Tool for Observational Cohort and Cross-Sectional Studies [49].

## Results

### Study characteristics

In this systematic review, we had a final sample of 15 studies examining Latina maternal depression in relation to child socioemotional development (**Table 2**). Study samples ranged from 26 to 1,600 mother-child participants per study (totaling 5,656 dyads). Participants in most studies (n = 14) came from low-income backgrounds, except one study using data from the Early Childhood Longitudinal Study–Birth Cohort (ECLS-B)—a nationally representative cohort of children born in 2001. Income levels for participants in the ECLS-B ranged from $\leq$ 20,0000 to $\geq$75,0000. Country of origin was reported in seven studies; six studies had mostly Mexican descent participants while one study was predominantly represented by participants of Puerto Rican descent [50]. Five studies employed samples from Head Start programs, which serves low-income families. Twelve of 15 studies were completed within the past ten years, and nine of those studies were published within the past five years.

Study designs included five cross-sectional studies and 10 longitudinal studies. Most studies measured child socioemotional outcomes through mother-reported scales, such as a) Child Behavior Checklist for ages 1 ½ to five (CBCL, n = 8), which measures internalizing and externalizing behavioral functioning [51]; and b) Infant Behavior Questionnaire (IBQ-R, n = 2) which assesses infant's regulatory capacity, negative effect, and extraversion [52]. Maternal depressive symptomology was measured by self-reported symptoms in all but one study, which used the DSM-IV diagnostic criteria for Major depressive disorder [53]. The most prevalent measure was the Center for Epidemiologic Studies-Depression Scale (CES-D, n = 9), which measures depression symptoms within the past week. Maternal acculturation was measured in nine studies, mostly through nativity (n = 6). Two studies measured maternal sensitivity: a mother's ability to perceive, interpret, and appropriately respond to child signals [54]. Measures of maternal sensitivity included The Three Bag Task or Two Bag Task [55], in which the dyad is asked to play with three toys during a 15-minute video-recorded session; trained coders later assess parent behavior.

### Data synthesis

We grouped key findings from selected studies into three categories based on individual, interpersonal, and contextual pathways linking maternal depression to child development (**Fig 3**): 1) child's socioemotional outcomes in relation to chronological developmental windows and the course and timing of maternal depression (prenatal, infancy, toddlerhood, preschool, and infancy through early childhood); 2) associations between maternal depression and maternal

**Table 1. Quality assessment tool for observational cohort and cross-sectional studies.**

| | Aisenberg, (2001) | Burtchen et al., (2013) | De Leon Siantz et al., (2010) | Doudna, (2016) | Fuller et al., (2018) | Huang et al., (2012) | La Roche et al., (1995) | Luecken et al., (2015) | Lequerica et al., (1995) | Martinez (2014) | Mennen et al., (2015) | Palermo et al., (2018) | Somers et al., (2018) | Waters et al., (2015) | Westbrook & Harden, (2010) |
|---|---|---|---|---|---|---|---|---|---|---|---|---|---|---|---|
| Was the research question or objective clearly stated? | Y | Y | Y | Y | Y | Y | Y | Y | Y | Y | Y | Y | Y | Y | Y |
| Was the study population clearly specified and defined? | Y | Y | Y | Y | Y | Y | Y | Y | Y | Y | Y | Y | Y | Y | Y |
| Was the participation rate of eligible persons at least 50%? | N | NR | NR | N | NR | Y | Y | Y | Y | Y | Y | NR | NR | NR | Y |
| Were all the subjects selected or recruited from the same or similar populations (including the same time period)? Were inclusion and exclusion criteria for being in the study pre-specified and applied uniformly to all participants? | Y | Y | Y | Y | Y | Y | Y | Y | Y | Y | Y | Y | Y | Y | Y |
| Was a sample size justification, power description, or variance and effect estimates provided? | N | N | N | N | N | N | N | N | N | Y | N | N | N | N | N |
| For the analyses in this paper, were the exposure(s) of interest measured prior to the outcome(s) being measured? | N | N | N | N | Y | Y | Y | Y | N | Y | Y | Y | Y | Y | Y |
| Was the timeframe sufficient so that one could reasonably expect to see an association between exposure and outcome if it existed? | N | N | N | N | Y | Y | Y | Y | N | Y | Y | Y | Y | Y | Y |
| For exposures that can vary in amount or level, did the study examine different levels of the exposure as related to the outcome (e.g., categories of exposure, or exposure measured as continuous variable)? | Y | Y | N | N | N | Y | N | N | N | Y | N | N | Y | Y | Y |
| Were the exposure measures (independent variables) clearly defined, valid, reliable, and implemented consistently across all study participants? | Y | Y | Y | Y | Y | Y | Y | Y | Y | Y | Y | Y | Y | Y | Y |
| Was the exposure(s) assessed more than once over time? | N | N | N | N | N | Y | Y | Y | N | N | Y | N | Y | Y | N |
| Were the outcome measures (dependent variables) clearly defined, valid, reliable, and implemented consistently across all study participants? | Y | Y | Y | Y | Y | Y | Y | Y | Y | Y | Y | Y | Y | Y | Y |
| Were the outcome assessors blinded to the exposure status of participants? | NR | NR | NR | NR | NR | NR | Y | NR | N | NR | NR | NR | NR | NR | NR |
| Was loss to follow-up after baseline 20% or less? | Y | Y | Y | Y | N | Y | Y | Y | Y | N | N | N | N | N | Y |
| Were key potential confounding variables measured and adjusted statistically for their impact on the relationship between exposure(s) and outcome(s)? | N | N | Y | N | Y | Y | N | Y | N | Y | Y | Y | Y | Y | Y |

Y = Yes, N = No, NR = Not reported

**Table 2. Summary of studies included in the review (n = 15, sample size = 5,656).**

| Study | Purpose | Study Design | Sample/Setting | Measure of Maternal Depression and data collection points | Measure of Socio-emotional outcome and data collection points | Country of origin and any acculturation data | Analysis and covariates | Results |
|---|---|---|---|---|---|---|---|---|
| | | | | **Prenatal and Infancy** | | | | |
| Burchen et al., (2013) | To examine the relationship between maternal major depression and infant social withdrawal behavior | Cross-sectional: infant's 6- month primary care visit. Exposure to clinical depression: (mean = 4 months prenatal, 6 months postnatal) | n = 155 (93%) Low-income predominantly (87%) Latina mother-child dyads from a subsample of a research project examining perinatal mood disorders and infant development. Maternal age: >18 M = 28 SD = 6.2. 6-month old full-term infants with no medical disorder or physical complaints at primary care routine visits | DSM-IV diagnostic criteria for Major Depression Disorder at 6 months after delivery: standardized psychiatric interview by a board-certified bilingual psychiatrist | Infant social withdrawal behavior at 6-months: ADBB (scale modified with author's permission) | Country of origin not reported maternal time in U.S. (M = 8.2 years) | Analysis: chi-square test or independent samples $t$ test. Covariates: potential confounding variables (i.e. maternal age, mode of delivery, number of children, child gender, tobacco, alcohol, and drug use) measured and deemed non-confounding | Infants with mothers diagnosed with major depression scored significantly higher on infant withdrawal behaviors than infants of mothers without depression. (6.1 vs 3.34, $p \leq .001$). Infants with depressed mothers scored significantly higher on intrapersonal and interpersonal social withdrawal behaviors: facial expression, vocalizations, relationship, attraction, eye contact, and self-stimulation. Significantly negatively correlated with maternal major depression: parents living together, father involved in childcare, emotional support, food security |
| Fuller et al., (2018) | To examine the relationship between prenatal material hardship with infant temperament at 10 months; prenatal depression used as a moderator | Longitudinal: 28-32-weeks gestation and 10 months of age | n = 412 Low-income Hispanic pregnant mothers enrolled in an obesity prevention program (Starting Early) uncomplicated singleton pregnancies at primary care clinics. Maternal age: >18 | PHQ-9 prenatally: 28-32-week gestation | Infant temperament (orienting/regulatory capacity, negative effect, surgency / extraversion) at 10 months: IBQ, Very Short Form | U.S. or foreign-born mothers. Country of origin not reported. U.S. born mothers had infants with higher negativity ratings (p = .01) unrelated to depressive symptoms | Analysis: linear regression. Covariates: marital status, immigration status, education, parity, infant sex, infant birth weight, intervention group, prenatal depression (used as either covariate or moderator), PPD at 3 months | Prenatal depressive symptoms significantly moderated the relationship between neighborhood stress and orienting/regulatory capacity scores, (standardized ß = -0.28, ß = -0.86; SE, 0.26; 95% CI, -1.37 to -0.34) |
| Luecken et al., (2015) | (1) Examine the relationship between prenatal maternal depressive symptoms and infant temperamental negativity as predictors of infant cortisol response (2) Explore the interaction between maternal depressive symptoms and infant negativity | Longitudinal: (26 to 38-week gestation), 6 and 12 weeks postpartum | n = 322 Mexican-American low-income (below $25,000) mother-infant dyads with healthy singleton pregnancies at hospital-based prenatal clinic. Maternal age: M = 27.8 SD = 6.5 18–42 | 10-item EPDS≥13 prenatally (26 to 38- week gestation). 9 and 12 weeks post-partum | Infant temperament negativity at 6 weeks: infant negativity subscale of IBQ-R. Infant dysregulation: saliva cortisol at 12 weeks: AUCg—prior to task(T0), at 0 (T1), 20 (T2), and 40 (T3) minutes after task complete | 86% born in Mexico; 14% U. S. born; 82% mostly Spanish speaking. Acculturation measured (as potential covariate) by ARSMA II. Mexican-born mothers reported decreased infant negativity compared to U.S.-born mothers(p < .01), significantly unrelated to other measured variables in study | Analysis: regression analyses (not specified). Covariates: time of day, postpartum: maternal AUCg, depressive symptoms, mood. Models analyzed with and without covariates | Higher prenatal maternal depressive symptoms correlated with elevated cortisol measures for 12-week old infants with high temperamental negativity but lower cortisol for infants with low negativity, (unstandardized estimate for the interaction = .019, SE = .007, 95% CI [.005, .033], p = .008). The effect of low social support followed a similar trend. Neither low prenatal depressive symptoms nor high prenatal social support predicted infant cortisol outcomes. Bi-directionally, higher infant negativity at 6 weeks predicted increased maternal depressive symptoms at 12 weeks (unstandardized ß = 1.12, t (297) = 3.29, p = .001; model R² = .18). |
| | | | | **Toddlerhood** | | | | |

*(Continued)*

**Table 2.** (Continued)

| Study | Purpose | Study Design | Sample/Setting | Measure of Maternal Depression and data collection points | Measure of Socio-emotional outcome and data collection points | Country of origin and any acculturation data | Analysis and covariates | Results |
|---|---|---|---|---|---|---|---|---|
| Huang et al., (2012) | To examine the relationship between maternal depression, maternal sensitivity and child attachment | Longitudinal: 9 and 24 months of age | n = 1600 Mother-child dyads from the Early Childhood Longitudinal Study–Birth Cohort | Modified CES-D at 9 months CIDI Short Form at 24 months | Child attachment measured at 24 months: TAS-45 | Country of origin not reported U.S. Born Hispanic: n = 750 Foreign Born Hispanic: n = 850 | Analysis: logistic regression and ANOVA Covariates: household income, maternal education, childcare arrangement | Chronic maternal depressive symptoms in Hispanic mothers posed the highest odds ratio for child insecure attachment (OR = 8.12, 95% CI = 1.07–61.68, p = .04). |
| | | | Maternal Age, U.S.-born: M = 26 15–40+ | Maternal sensitivity at 24 months: Two Bag Task | | Compared with U.S. born Hispanic women, foreign-born mothers are significantly less likely to have an insecurely attached child (OR = .69, 95% CI = .51-.95, p = .02). | | "Maternal sensitivity" did not significantly mediate the relationship between maternal depression and insecure attachment (OR = .85, 95% CI = .71–1.03, p = .10). |
| | | | Maternal age, foreign-born: M = 27.8 15–40+ | | | | | The most common depressive pattern in Hispanic women, later onset at 24 months, predicted the least likelihood for insecure attachment at 24 months (OR = .32, 95% CI = .12-.88, p = .03). |
| | | | Child age at Time 1: 9 months Child age at Time 2: 24 months | | | | | |
| Martinez, (2014) | To examine the relationship between maternal depression and child behaviors (aggression, compliance, and negative emotionality) | Longitudinal: Baseline and 6-month follow up | n = 47 Latinas with limited English proficiency, low-income and a positive depression screen; mother-child dyads from a larger Interpersonal Psychotherapy study enrolled in Early Head Start programs | CES-D≥16 at baseline | Child behavior (aggression, compliance, and negative emotionality) at baseline and at 6-month follow-up: CBCL and ITSEA-R | Country of origin not reported Years in U.S: M = 5.3 years, SD = 5.9 Acculturation: SASH | Analysis: descriptive statistics, zero-order correlations, and hierarchal multiple regression analyses Covariates: child gender, child age, treatment condition, and child behavior at baseline | Maternal depressive symptoms at baseline significantly predicted child negative emotionality 6 months later: [F (5, 23) = 3.56, p < .05, R2 = .44, R2Δ=.09, p < .10] |
| | | | Maternal age: M = 27.13 SD = 5.6 | | | | | Severity of maternal depressive symptoms x child negative emotionality at baseline improved models predicting child negative emotionality 6 months later: F (5, 22) = 4.10, p < .01, R2 = .53, R2Δ = .09, p = .051 |
| | | | Child age: M = 23.1 months SD = 8.7 | | | | | Maternal depressive symptoms significantly moderated the relationship between negative emotionality at baseline and at 6-month follow-up: (B = -.02, p < .05) |
| | | | | | | | | Depression severity significantly moderated these associations: Low: t (22) = 3.74, p = .001 Average: t (22) = 3.47, p = .002 High: t (22) = 1.58, n.s. |
| | | | | | | | | Child aggression and compliance were not significantly associated with maternal depression |
| | | | | | **Preschool** | | | |
| Aisenberg, (2001) | To examine the psychological and behavioral effects of exposure to community violence | Cross-Sectional: maternal and child measures at study entry | n = 31 Low-income Latina mother-child dyads enrolled in a Head Start program | "Maternal Distress Symptomology" at study entry: Adult PTSD: IES-R Depression and anxiety: BSI≥63 | Child behavioral functioning measured at study entry: CBCL | 80.6%: Born in Mexico 6.1%: Born in El Salvador 12.9%: U.S.-born mothers | Analysis: Univariate t tests without correction for multiple comparisons and bivariate correlations, multiple regressions Covariates: none reported | Maternal distress symptomatology did not moderate relationship between exposure to community violence and CBCL scores |
| | | | Maternal age: M = 28.74 SD = 4.77 21–43 | | | | | Maternal distress symptomatology significantly mediated the relationship between exposure to community violence and CBCL scores, β = .45, p < .05. |
| | | | Child age: 48–58 months M = 53.22 SD = 2.16 | | | | | Proportion of children with behavior problems based on CBCL≥60: 30%. |

(*Continued*)

**Table 2.** (Continued)

| Study | Purpose | Study Design | Sample/Setting | Measure of Maternal Depression and data collection points | Measure of Socio-emotional outcome and data collection points | Country of origin and any acculturation data | Analysis and covariates | Results |
|---|---|---|---|---|---|---|---|---|
| De Leon Siantz et al., (2010) | To examine the relationship between maternal functioning and child behavior problems | Cross-Sectional Prospective Design: maternal and child measures at study entry | n = 205<br><br>Latino children from "Migrant Head Start Programs",<br><br>Maternal age: M = 32 SD = 7.43 19–67<br><br>Child age: 36–72 months M = 51 SD = 6.24 | "Maternal Functioning" at study entry:<br><br>Depression: CES-D ≥16,<br><br>Maternal stress: FILE<br><br>Parenting Style: PARQ | Child behavioral functioning measured at study entry: CBCL | Acculturation measured by: HHANES and language preference 35.9% born in U.S, 61.4% born in Mexico | Analysis: step-wise regressions<br><br>Covariate: Maternal years in U.S.<br><br>How long child has been in Head Start could be confounding variable but not addressed | Internalizing problems significantly predicted by both maternal stress and depressive symptoms in both genders: (ß = .295, R² = .164, df = 2,145, F = 15.99, p = .000)<br><br>Maternal stress and depressive symptoms more likely to predict behavior problems in girls while maternal stress and harsher parenting style more likely to predict behavior problems in boys<br><br>Proportion of children with behavior problems based on CBCL 12% |
| Doudna, (2016) | To examine the relationship between household food insecurity, parenting alliance, and maternal depressive symptoms on child socioemotional outcomes within Latino families in rural America using the Family Stress Model | Cross-sectional: maternal and child measures at study entry | n = 99<br><br>Low-income Latina mothers living in rural communities enrolled in "Rural Families Speak about Health" project<br><br>Maternal age: >18 M = 32.36 SD = 7.87<br><br>Child age: 18–71 months | CES-D, short form ≥10 at study entry | Child behavioral functioning measured at study entry: CBCL | Not reported | Analysis: path analysis<br><br>Covariate: financial distress measured by PFW<br><br>No other potential confounding variables such as wide age range measured | Depressive symptoms were negatively associated with parenting alliance (ß = -2.7, p = .12)<br><br>Maternal Depressive symptoms did not significantly mediate the relationship between household food insecurity and child behavioral functioning |
| La Roche et al., (1995) | To examine the relationship between toddlers' behavioral difficulties, mothers' depression, self-efficacy, and social support | Longitudinal: Baseline and 3-month follow-up | n = 26<br><br>Low SES, Spanish-speaking Latina mother-child dyads attending a behavioral group in a community mental health center<br><br>Maternal age: M = 26.7 years 21–36<br><br>Child age: M = 34 months 24–60 | At baseline and 3-month follow-up:<br><br>Social Support: Norbeck Social Suport Network Scale<br><br>Depression: CES-D≥16<br><br>Self-Efficacy: The Maternal Efficacy Questionnaire adapted for toddlers | Preschool<br><br>Behavioral Checklist completed by both bilingual psychologists and mothers at baseline (independent variable) and 3-month follow-up (dependent variable) | Country of origin and acculturation data not reported | Analysis: Pearson correlations, multiple regression analyses<br><br>Covariates: none reported | Significant relationship between perceived social support at baseline and maternal depressive symptoms at 3-month follow-up, r = -.46 (p<0.05), -.51 (p < .05)<br><br>Maternal depressive symptoms at time one were not significantly associated with their toddlers' behavioral difficulties at 3-month follow-up. |
| Lequerica Et. al, (1995) | To examine mothers' concerns about their children's behaviors at home in relation to stressful family life events, maternal depression, methods of discipline, and demographic factors | Cross-sectional: maternal and child measures during out-patient pediatric clinic visit | n = 52<br><br>Low-income mother-child dyads seeking services at a pediatric outpatient clinic<br><br>Maternal age: Not reported<br><br>Child age: 24–36 months: 32.7% 36–48: 38.5% 48–60: 28.8% | At study entry:<br><br>Depression: subscale of Ilfeld's Psychiatric Symptom Index<br><br>Stressful life events: Social Readjustment Rating Scale by Holmes and Rahe,—shortened version | Child behavioral functioning measured at study entry: CBCL (shortened version with 65 questions) | Country of origin: Puerto Rico: 55.8% Santo Domingo/Central America: 34.6%<br><br>South America: 9.6%<br><br>Acculturation data not reported | Analysis: Pearson correlations, chi squares, and analysis of variance<br><br>Covariate: none reported | CBCL scores were not significantly related to family life stressful events or maternal depressive symptoms.<br><br>CBCL questions with higher frequencies than normal or psychiatrically referred non-Latino children:<br><br>Clings to adults: 80% of 4 to 5 year olds compared to 32% in Achenbach's normal<br><br>Unable to sit still/hyperactive: 90% of 4 to 5 year olds compared to 40% in Achenbach's normal<br><br>(Continued) |

**Table 2.** (Continued)

| Study | Purpose | Study Design | Sample/Setting | Measure of Maternal Depression and data collection points | Measure of Socio-emotional outcome and data collection points | Country of origin and any acculturation data | Analysis and covariates | Results |
|---|---|---|---|---|---|---|---|---|
| Mennen et al., (2015) | To examine the relationship between clinical maternal depression and children's progress during mental health treatment | Longitudinal: at study entry and at 6-month intervals up to 3 years | n = 147 Low-income predominantly Latina (94%) mother-child dyads. Child in or at high risk for child welfare services, diagnosed with emotional, behavioral or mental disorder, and enrolled in an inner-city mental health treatment program (Project ABC). Maternal age: M = 30 SD = 6.6. Child age: M = 34 months SD = 14.8, 58% boys | CES-D $\geq$16 at study entry and at 6-month intervals up to 3 years | Child behavioral functioning measured at study entry and at 6-month intervals up to 3 years: CBCL for children ages 1.5–5. Adaptive functioning measured at study entry and at 6-month intervals for up to three years: Vineland Screener | Country of origin and acculturation data not reported | Analysis: univariate statistics, growth curve modeling. Covariates: child sex and child welfare involvement. Study did not adjust for maternal depression, a potential confounding variable given self-reported child functioning measures | Significant associations between maternal depressive symptoms and child behavior problems at entry: higher internalizing scores (17.70 vs 14.23, $p$ = .018), higher externalizing scores (24.65 vs 21.45, $p$ = .044), higher total CBCL scores (65.09 vs 52.99, $p$ = .006), and lower socialization scores, (95.58 vs 103.36, $p$ = .022). Children's CBCL scores improved more slowly when their mother was depressed: Internalizing = 1.59, $p <$ .05; Externalizing = 17.75, $p<$ .05; Total = 4.66, $p <$ .05); daily living skills: time-by-depression interaction coefficient = 2.99, $p <$ .05). Children of mothers wither higher depressive symptoms: finished treatment with lower levels of behavioral functioning, received more services (96.3 vs. 61.4, $F$ = 4.18, $p$ = .043), and remained in treatment for longer (435 vs. 257 days, $F$ = 4.41, $p$ = .039). No significant associations with maternal depressive symptoms on child's communication, daily living skills, and motor skills |
| Palermo et al., (2018) | To examine the relationship between economic hardship during infancy, "maternal mental health problems", "maternal positive parenting behaviors", and Latino children's socio-behavioral difficulties and academic skills prior to kindergarten entry-using a culturally integrated Family Stress Model; do acculturation levels moderate the pattern of associations? | Longitudinal: children at 4, 14, 24 and 36 months, and half a year before kindergarten | n = 714 Early Head Start Research and Evaluation Project (EHSREP) Low-income Latina mothers. Maternal age: 84.6%$\geq$18 M = 24 SD = 6 13–43. Child age at study entry: M = 4 months | At 14 months, "maternal mental health problems" quantified by Depression: CES-D, Parenting Stress: PSI-SF, Perceived Control: Pearlin Mastery Scale. Maternal sensitivity at 14 months: Three Bag Task | Both measures within half a year before kindergarten: -Socio-behavioral health problems (SBHP): Problem Behavior scale. Subscales: aggressive behavioral scale, hyperactive behavior scale | 82% Mexican-American; 59% foreign-born; 9% Central American; 6% Puerto Rican. At 24 months old, mother's acculturation levels measured by generational status, English use preference, and proficiency: Multi-cultural Acculturation Scale, Picture Vocabulary subscale of the Woodstock-Johnson Achievement III. Positive association between maternal acculturation and positive parenting behaviors ($\beta$ = .22, SE = .06, $p <$ .001) | Analysis: Structural equation modeling. Covariates: - child's gender, maternal education, family structure. -At 14 months: hyperactivity measured by 1 item from the Bayley Scales of Infant Development II | Positive association between economic hardship and maternal mental health problems ($\beta$ = .50, SE = .08, $p <$ .001). Negative association between maternal mental health problems and maternal positive parenting behaviors ($\beta$ = -.16, SE = .06, $p$ = .008). Maternal mental health problems and maternal positive parenting behaviors did not significantly mediate the association between economic hardship and children's socio-behavioral problems ($\beta$ = -.02, SE = .01, $p$ = .061). Maternal positive parenting behaviors significantly mediated the association between maternal mental health problems and social behavioral problems ($\beta$ = .03, SE = .02, $p$ = .047). Relationship patterns did not vary by acculturation levels |

(*Continued*)

**Table 2.** (Continued)

| Study | Purpose | Study Design | Sample/Setting | Measure of Maternal Depression and data collection points | Measure of Socio-emotional outcome and data collection points | Country of origin and any acculturation data | Analysis and covariates | Results |
|---|---|---|---|---|---|---|---|---|
| Westbrook & Harden, (2010) | To examine the relationship of proximal risk factors (family structure and maternal depression) and distal risk factors (community violence) on parenting style and subsequent preschool children's socioemotional outcomes in an ethnically and racially stratified sample using the Family Stress Model | Longitudinal: Data Analysis from the FACES 2000 cohort from Fall 2000 (Time 1) to Spring 2001 (Time 2) for families with children aged 3–4 years old | n = 425 FACES 2000 (The Family and Child Experiences) study cohort derived from a Nationally representative sample of Head Start programs intended to measure program quality through socioemotional and academic outcomes Maternal age: >18 M not provided Child age: M = 48.3 months SD = 6.4 at baseline (Time 1) | Modified CES-D used to measure depressive symptomology at baseline (Time 1) | Measured at Fall 2000 and Spring 2001: Social skills measured by selected items from the Personal Maturity Scale and the Social Skills Rating System Behavior Problems measured by selected items on Personal Maturity Scale, CBCL, Teacher Report and the Behavior Problems Index | Country of origin and acculturation data not reported | Analysis: structural equation model Covariates: maternal age, maternal education, maternal employment, family poverty status, child age, child gender, baseline social-emotional functioning | No significant direct associations between individual study variables (violence exposure, family structure, maternal depression, parenting style) on child socio-emotional outcomes Cumulative effect of variables explained 36% of the European-American model and 22% of the Latino model |
| **Infancy through Early Childhood** | | | | | | | | |
| Somers et al., (2018) | (1) To examine the relationship between postpartum maternal depression and children's biological sensitivity to behavior problems via children's dysregulation and (2) To explore sex differences in behavior problems due to PPD | Longitudinal: Every 3 weeks postpartum from 6 to 24 weeks, at 24 months, and at 36 months | n = 322 Low-income Mexican-American mother-child dyads from a larger study (Las Madres Nuevas) A healthy singleton infant Maternal age: >18 M = 27.8 SD = 6.5 | 10-item EPDS every 3 weeks from 6 weeks to 6 months ≥13 CES-D at 36 months (as covariate) ≥16 | Child dysregulation at 24 months: validated dysregulation coding system Child Behavior Problems at 36 months: 113-item CBCL | 86% born in Mexico; 13.7% born in U.S Children of U.S.-born mothers: -higher RSA, p = .003 -more behavior problems, p = .001 country of birth predicted behavior problems (B = -14.350, SE(B) = 4.712, p = 0.002 CI (-23.59, -5.12) | Analysis: structural equation model Covariates: maternal country of origin, maternal age, number of biological children, and concurrent depressive symptoms | Total child behavior problems were significantly predicted by PPD symptoms x infant RSA: (B = -0.221, SE(B) = 0.069, p = 0.001) Low infant RSA x PPD symptoms conferred the most susceptibility (B = 0.190, SE(B) = 0.097, p = 0.05) PPD symptoms x infant RSA predicted: -internalizing behavior problems: (B = -0.083, SE (B) = 0.022, p ≤ .001) -externalizing behavior problems, (B = -0.068, SE (B) = 0.025, p = 0.006) Higher maternal 36 month depressive symptoms associations: -lower infant RSA, r = -0.177, p = 0.22 -greater PPD symptoms, r = .349, p ≤ .001 -more behavior problems, r = 0.482, p ≤ .001 Higher depressive symptomology = more significant effect on child behavior problems Child dysregulation was not significantly predicted by any of the study variables No significant sex differences |

(*Continued*)

Table 2. (Continued)

| Study | Purpose | Study Design | Sample/Setting | Measure of Maternal Depression and data collection points | Measure of Socio-emotional outcome and data collection points | Country of origin and any acculturation data | Analysis and covariates | Results |
|---|---|---|---|---|---|---|---|---|
| Waters et al., (2015) | Does chronic maternal depression and chronic overcrowding moderate the relationship between ANS (automatic nervous system) reactivity at 6 months and externalizing behaviors at 7 years old? | Longitudinal: 6 months, 1 year, 3.5 years, and 7 years of age | n = 99<br>The Center for the Health Assessment of Mothers and Children of Salinas birth cohort (CHAMACOS)<br><br>Mexican-American low-income mother-child dyads<br><br>Maternal age:<br>M = 26.49<br>SD = 4.75<br>18–42 | CES-D ≥ 16 at 1-year old and 3.5-year old visits | Child externalizing problems at 7-years old: BASC-2 | 92% women born in Mexico, 94% only/mostly Spanish speaking | Analysis: multiple linear regression<br><br>Covariates: child sex, maternal years in the U.S. | In children with low resting sinus arrhythmia (RSA) reactivity, chronic maternal depressive symptoms predicted highest levels of externalizing difficulties at 7 years, t (74) 5 3.98, p < .001, 95% CI [10.04, 30.17].<br><br>In children with high RSA reactivity, chronic maternal depressive symptoms predicted the lowest levels of externalizing difficulties at 7 years<br><br>No association between children's automatic nervous system reactivity scores and maternal depressive symptoms<br><br>No associations between overcrowded housing and externalizing behaviors |

ADBB: Alarm Distress Baby Scale; ARSM II: Acculturation Rating Scale for Mexican Americans; BASC-2: Behavior Assessment System for Children 2; BSI: Brief Symptom Inventory; BSID-II: Bayley Scales of Infant Development, Second Edition; CES-D: Center for Epidemiologic Studies- Depression Scale; CBCL: Child Behavior Checklist; CIDI: Composite International Diagnostic Interview; IBQ-R: Infant Behavior Questionnaire Revised; IES-R: Impact of Events Scale–Revised; FILE: Family Inventory of Life Events and Changes Scale; ITSEA-R: Infant Toddler Social Emotional Assessment; HHANES: Hispanic Health and Nutrition Examination Survey; MDI Bayley: Bayley II Mental Developmental Index; PARQ: Parental Acceptance/Rejection Questionnaire; PFW: Personal Financial Wellness Scale; PHQ-9: Patient Health Questionnaire-9; PSI-SF: The Parenting Stress Index–Short Form; SASH: Short Acculturation Scale for Hispanics; TAS-45: Toddler Attachment Sort-45 Instrument;.

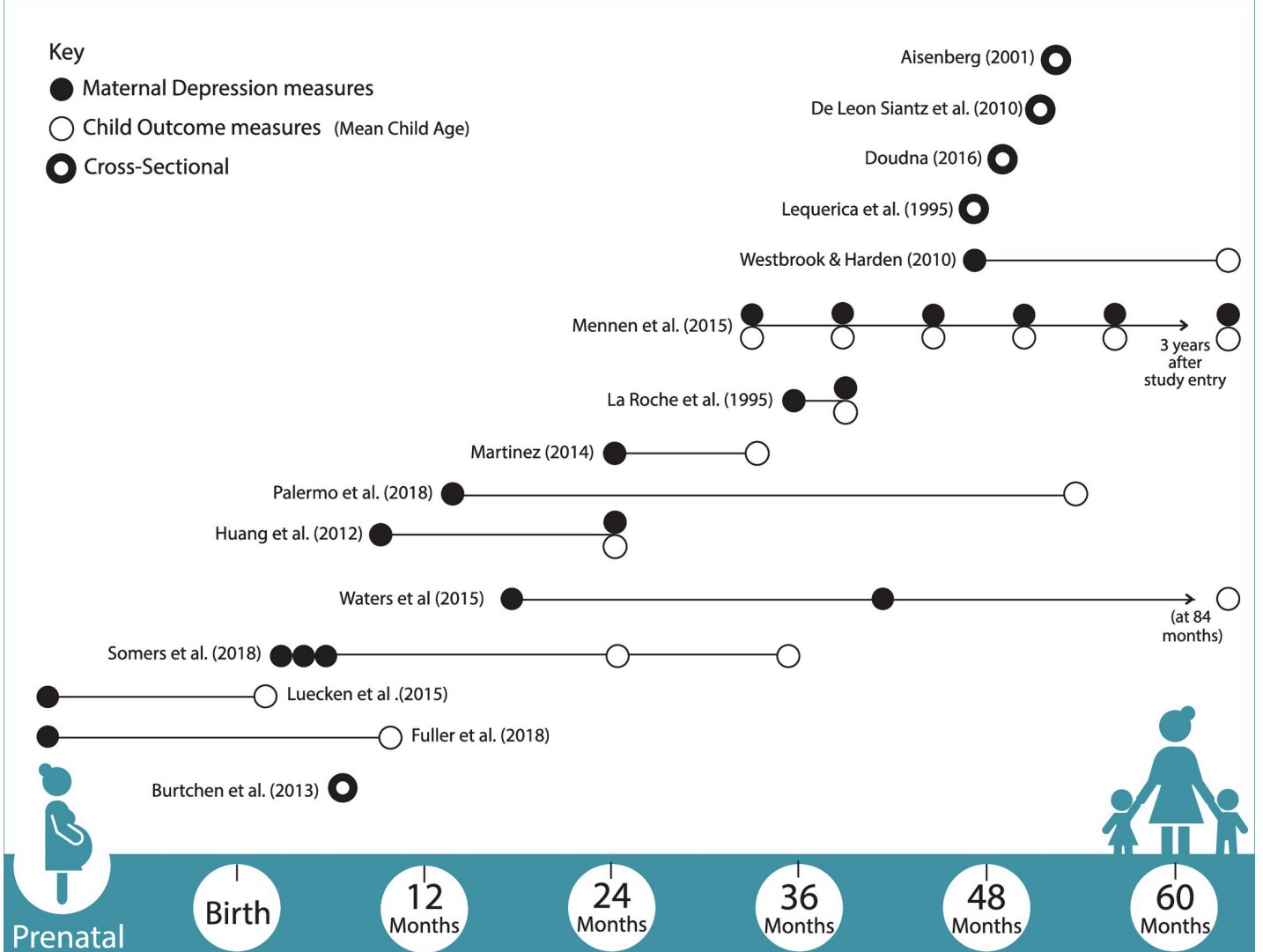

**Fig 3. Timeline of studies reviewed (n = 15) based on maternal depression and child outcome assessments.**

sensitivity in relation to child socioemotional outcomes; and 3) contextual stressors culturally relevant to Latina mothers and their young children living in poverty (economic and sociocultural stressors).

## Socioemotional outcomes

**Prenatal.** Studies (n = 2) conducted in the prenatal period found a significant relationship between prenatal depressive symptoms and infant self-regulation [56,57]. In Fuller et al.'s study, higher prenatal maternal depressive symptoms on a continuum (Patient Health Questionnaire-9, score range 5–27) positively moderated the relationship between neighborhood stress and infant self-regulatory difficulties (IBQ-R). Luecken et al.'s study measured the relationship between infant self-regulation (cortisol responses) and prenatal depressive symptoms (Edinburgh Postnatal Depression Scale ≥13). Both prenatal depressive symptoms and perceived lack of social support predicted disrupted infant self-regulatory processes as indicated

through dysfunctional cortisol responses: heightened cortisol output for 12-week old infants with high temperamental negativity but blunted cortisol for infants with low negativity [57]. Both studies co-varied for postpartum depressive symptoms, suggesting that dysfunctional self-regulatory processes were associated with the prenatal rather than postnatal psychosocial environment.

**Infancy.** Studies conducted during infancy (n = 2) revealed significant correlations between maternal depression and adverse infant socioemotional outcomes. Six-month old infants whose mothers were diagnosed with major depression (based on the DSM-IV criteria) scored significantly higher than infants with non-depressed mothers on both intrapersonal and interpersonal social withdrawal behaviors (Alarm Distress Baby Scale): facial expression, vocalizations, relationship, attraction, eye contact, and self-stimulation [53]. Bidirectional interpersonal associations, implying disrupted social synchrony, were noted in a separate study: higher infant negativity (IBQ-R) at six weeks was associated with increased maternal depressive symptoms (EPDS > 13) at 12 weeks; moreover, increased maternal depressive symptoms were associated with decreased social support [57]. Each study measuring infant outcomes (n = 4) yielded a significant association between maternal depressive symptoms and adverse infant outcomes regardless of study design, analytical analysis, child outcome measures, or maternal depression measures.

**Toddlerhood.** Two studies focused on the relationship between maternal depressive symptoms and child outcomes in toddlers aged 12 to 36 months. Martinez examined the relationship between maternal depressive symptoms (CES-D $\geq$ 16) and child aggression, compliance, and negative emotionality (CBCL and Infant Toddler Social Emotional Assessment) [58]. Mild and moderate depressive symptoms at 24 months significantly predicted increased negative infant emotionality six months later [58]. Huang et al. examined the relationship between maternal depressive symptoms (CES-D $\geq$ 16 at nine months, Composite International Diagnostic Interview Short Form Instrument at 24 months), maternal sensitivity, and toddler child attachment (Toddler Attachment Sort-45 Instrument). Analyses stratified by ethnicity revealed that chronic maternal depression in Latina mothers posed a higher odds ratio for child insecure attachment at 24 months than any other ethnicity (Odds Ratio = 8.12, 95% Confidence Interval = 1.07–61.68, p = .04) [2]; however, the significance of this association disappeared when maternal sensitivity was added into the model [2]. As such, both studies found significant relationships between toddler socioemotional outcomes and the chronicity [2] or severity [58] of maternal depressive symptoms.

**Preschool.** Four studies examined the correlation between maternal depressive symptoms and behavioral functioning (CBCL) in preschoolers. Neither La Roche et al. or Lequericia and Hermosa found significant associations between maternal depressive symptoms (CES-D $\geq$ 16 or subscale of Ilfield's Psychiatric Symptom Index) and preschool behavioral functioning [50,59]. However, neither study adjusted for potential confounding variables (e.g., maternal demographic characteristics, child's age), which could have obscured possible associations. Only when maternal depressive symptoms (CES-D $\geq$ 16) were combined with measures of stress did significant associations emerge between maternal depression and child behaviors: maternal depressive symptoms in conjunction with stress was significantly positively associated with child internalizing problems [60]. Two of these studies were cross-sectional [50,60], and one study spanned a three-month time-frame [59], providing a limited perspective into the trajectory of these associations.

During children's mental health treatment spanning three years, Mennen et al. found significant associations between maternal depressive symptoms (CES-D $\geq$ 16) and preschoolers' behavioral functioning: externalizing and internalizing behaviors (CBLC) and socialization (Vineland Screener) [61]. Compared to children whose mothers were not depressed, children

of mothers with depression began treatment with higher levels of internalizing and externalizing behaviors and lower socialization scores, progressed more slowly during treatment despite receiving more mental health services, and finished treatment with lower levels of behavioral functioning [61]. Unlike the consistently significant associations found between maternal depression and adverse infant outcomes, significant correlations during the preschool years emerged only when 1) the study design and analysis captured developmental changes over time or when 2) maternal depressive symptoms were combined with a measurement of stress.

**Infancy through early childhood.** In longitudinal studies spanning infancy through early childhood (n = 2), the interaction between maternal depressive symptoms and an infant's regulatory capacity was found to predict child behavioral functioning through the preschool years [62] and middle childhood [63]. Infants' resting respiratory sinus arrhythmia (RSA) [62] and RSA reactivity [63] were measured as a physiological index for vagal tone and subsequent underlying regulatory capacity. The interaction between low infant RSA reactivity and exposure to maternal depressive symptoms (CES-D $\geq$ 16) at one and three-and-a-half years old predicted significantly greater externalizing difficulties (Behavior Assessment System for Children) at 7-years-old [63]. Likewise, the interaction between low infant resting RSA and exposure to depressive symptoms (EPDS $\geq$ 13) from three weeks to six months old predicted greater internalizing and externalizing child behavior problems (CBCL) at 36 months [62]. Additionally, higher maternal depressive symptoms (CES-D $\geq$ 16) at 36 months yielded other significant correlations: maternal depressive symptoms (EPDS $\geq$ 13) at six months and more child behavior problems [62]. In summary, the interaction between infant self-regulatory processes and the chronicity of maternal depressive symptoms predicted greater child behavioral difficulties [62,63]. Measurements assessing exposure to maternal depressive symptoms across multiple points in the child's life strengthened both studies.

## Maternal sensitivity

Two studies examined the relationship between maternal depressive symptoms, maternal sensitivity, and acculturation during toddlerhood and preschool [2,64]. In the ECLS-B's full sample representing the general population, maternal depressive symptoms measured at nine months (CES-D $\geq$ 16) and 24 months (Composite International Diagnostic Interview Short Form Instrument) were inversely associated with maternal sensitivity scores [2]. However, in analyses stratified by ethnicity, patterns of depression at nine months and 24 months (e.g., remittent, late onset, chronic) were not significantly associated with maternal sensitivity in Latina mothers [2]. In contrast, Palermo et al. found a significant inverse association between maternal sensitivity and maternal mental health problems–a variable combining maternal depressive symptoms (CES-D), parenting stress, and self-efficacy measures. In both studies, more acculturated mothers scored significantly higher on maternal sensitivity measures than less acculturated mothers. In fact, maternal acculturation (as measured by either primary language use, nativity, or a combination of generational status and language proficiency), not maternal depressive symptoms, was most consistently and significantly associated with maternal sensitivity scores [2,64].

Maternal sensitivity was also tested as a mediator between maternal depressive symptoms and child outcomes. Combining stress and depression measures into one variable again yielded the stronger association: maternal sensitivity significantly mediated the association between maternal mental health problems and child behavioral functioning (Problem Behavior Scale) in preschoolers [64]. Although Huang et al. found that maternal sensitivity mediated the associations between maternal depressive symptoms and child attachment in toddlers, the mediation was not significant in Latina mother-child dyads–a finding that differed from the

ECLS-B's full sample [2]. In summary, unless maternal depressive symptoms were combined with maternal stress, maternal sensitivity scores were not correlated with maternal depression and did not function as a significant mediator between maternal depressive symptoms and child outcomes. Both studies assessed and adjusted for demographic moderators (e.g., socio-economic status, maternal education).

## Economic and sociocultural stressors

**Economic stressors.** Studies using the FSM (n = 3) during early childhood found consistent positive correlations between maternal depressive symptoms and economic stressors: food insecurity, community violence, and economic hardship [64–66]. Maternal depressive symptoms were then tested as a mediator between these poverty-related stressors and child outcomes. In a cross-sectional study examining Latino families in rural America across 13 states, maternal depressive symptoms (CES-D short form ≥ 10) were not found to significantly mediate the relationship between household food insecurity and child behavioral functioning (CBCL) [65]. The study was limited by its cross-sectional nature and the potentially confounding child age range (one-and-a-half to five years old, mean = 42 months). However, a longitudinal study examining children from a nationally representative sample of Head Start programs across one school year also found statistically non-significant effects: child behavioral functioning (Personal Maturity Scale, CBCL, Teacher Report, and the Behavior Problems Index) was not significantly associated with individual study variables: maternal depressive symptoms (CES-D short form ≥ 10), violence exposure, family structure, or parenting style [66]. Furthermore, maternal mental health problems (CES-D ≥ 16, parenting stress, self-efficacy) and maternal sensitivity did not mediate the association between economic hardship and child behavioral functioning (Problem Behavior Scale) in a longer longitudinal study spanning 14 months until kindergarten entry [64].

Although the FSM studies provided no support for the mediating role of maternal depressive symptoms between poverty-related stressors and child outcomes, Aisenberg's cross-sectional study found that maternal distress symptomatology, as measured by depressive symptoms (BSI ≥ 63), anxiety, and adult post-traumatic stress disorder, significantly mediated the relationship between exposure to community violence and externalizing and internalizing behaviors (CBCL) in preschool children [67].

Two studies examined maternal depressive symptoms as a moderator between economic stressors and child outcomes. Waters et al. tested both chronic overcrowded housing and chronic maternal depressive symptoms (CES-D ≥ 16) as potential moderators between child RSA reactivity and child externalizing behavioral difficulties (BASC-2) from infancy through middle childhood [63]. Only maternal depressive symptoms functioned as a significant moderator, indicating that maternal depression posed a stronger risk for children than overcrowded housing in a sample of Mexican-American dyads [63]. Fuller et al. tested maternal depressive symptoms on a continuum (PHQ-9, score range 5–27) as a moderator between infant temperament (IBQ-R) and prenatal material hardship (food insecurity, housing disrepair, financial difficulty, neighborhood stress) [56]. Prenatal depressive symptoms significantly moderated the relationship between neighborhood stress and low infant regulatory capacity [56].

**Acculturation.** Two studies examined the relationship between maternal depressive symptoms and child socioemotional outcomes in context of acculturation (i.e., cultural adaptation) [2,64]. A culturally integrated FSM tested whether acculturation levels moderated the pattern of associations between maternal mental health problems (CES-D, parenting stress, self-efficacy) and preschool behavioral functioning (Problem Behavior Scale) [64]. Palermo

et al. found significantly higher measures of maternal sensitivity in more acculturated mothers compared to less acculturated mothers, but adding maternal acculturation measures to models did not moderate the associations between maternal mental health problems and child outcomes. Nor did maternal nativity status moderate the associations among maternal depressive symptoms (CES-D ≥ 16), maternal sensitivity, and child attachment [2].

Despite the lack of studies explicitly examining the effects of maternal acculturation on the relationship between maternal depression and child development, a significant pattern repeatedly emerged across studies. Children of U.S.-born Latina mothers had consistently poorer socioemotional outcomes than children of foreign-born Latina mothers across domains and developmental windows: higher infant negativity in two studies [56,57]; higher rates of insecure attachment in toddlers, including almost double the rate (20.7% vs 11.5%) of disorganized attachment, the most maladaptive style [2]; and higher externalizing and internalizing behavioral problems in preschoolers [62]. Higher rates of insecure attachment in toddlers (49.4% vs 39.6%) occurred *despite* higher maternal sensitivity scores, family incomes, and maternal education levels in U.S-born Latina mothers from the ECLS-B [2]. Furthermore, higher child internalizing and externalizing behavior problems in preschoolers occurred *despite* higher resting RSA levels, a measure found to protect children significantly from socioemotional difficulties in the full sample of Mexican-American children [62]. Potential links between maternal nativity status *and* maternal depression in relation to child outcomes remained predominantly unexplored, leaving a gap in our understanding.

## Discussion

This systematic review examined the relationship between maternal depression in Latina mothers and their children's socioemotional outcomes from birth to five years of age. Fifteen studies with a total of n = 5,656 mother-child participants met our inclusion criteria. Of the eleven studies published within the past ten years, seven were published in the past five years, indicating that the relationship between maternal depression in Latinas and child development has only recently begun to receive scholarly attention.

Consistent with previous meta-analyses conducted on European-White mother-child dyads, the preschool years were the least likely to yield statistically significant inverse associations between maternal depression and child well-being; infancy had the strongest inverse correlations, suggesting that younger children may be more vulnerable to the influences of maternal depression [1,8]. Chronicity and severity of maternal depression also predicted child outcomes; more frequent exposure to maternal depression across infancy and early childhood was linked to increased child maladjustment by early and middle childhood. We also found some evidence for the moderating [56,63] and mediating role [67] of maternal depression between contextual stressors and child outcomes. Maternal stress in conjunction with depression heightened associations between maternal depression and child maladjustment [60,64,67]. In contrast, maternal sensitivity less consistently affected the relationship between maternal depression and child outcomes [2,64]. One of the most consistent patterns emerged between maternal nativity and child socioemotional development: children of U.S.-born Latina mothers had significantly poorer developmental outcomes than children of foreign-born Latina mothers across socioemotional domains and throughout early developmental windows—from infant negativity to toddler attachment and preschool behavioral functioning [2,56,57,62].

### Maternal sensitivity

Maternal sensitivity scores were more consistently correlated with maternal acculturation than maternal depression, whereby high levels of maternal acculturation were linked to high levels

of maternal sensitivity scores—thus raising the question "Are maternal sensitivity measures more closely linked to cultural assimilation than maternal processes that protect children?" Although previous research has supported maternal sensitivity as a protective cross-cultural construct critical for healthy child developmental outcomes [68], some evidence asserts that measures used to operationalize this complex construct may be culture-specific [69]. In fact, the Nursing Child Assessment Teaching Scale, used to measure maternal sensitivity during infancy in the ECLS-B, may detect maternal knowledge of early child development more sensitively than either maternal depression or self-efficacy [70]. Although positive correlations between maternal sensitivity and preschooler's behavioral functioning suggest that maternal sensitivity measures may at least partially apply to Latina mothers in relation to child well-being [64], measures are likely not capturing key maternal processes specific to less acculturated Latina mothers; these processes warrant further exploration. Moreover, considering that increased maternal sensitivity scores in more acculturated Latina mothers did not moderate the relationship between maternal depression and child outcomes in toddlers or preschoolers [2,64], unexplored interpersonal or contextual factors in relation to maternal depression may play a superseding role in determining child well-being.

## Economic and sociocultural stressors

Depression has long been recognized as an interdependent phenomenon inextricably linked to interpersonal and contextual factors [71]. As such, economic stressors and lack of social support were significantly associated with maternal depression in our studies. We also found some evidence for the mediating [67] and moderating [56,63] role of maternal depression between poverty-related stressors and child socioemotional outcomes. Our review found no support for using the FSM during the preschool years, in contrast to FSM studies conducted during middle childhood and adolescence within the broader literature [31,32,72]. Perhaps the cumulative effects of chronic stressors are not measurable until later in childhood. The preschool years, however, were the least likely to yield significant associations between maternal depression and child development in our studies unless maternal depression was combined with a measure of stress. Elevated scores on both maternal depression and stress measures may reflect more chronic depressive symptoms [73], a pattern of depression associated with greater risk for child maladjustment in our longitudinal studies [2,62,63]. Moreover, some empirical evidence suggests that maternal stress (e.g., parenting stress, limited social support, marital conflict) may mediate the relationship between maternal depression and behavior problems in toddlers [74] and preschoolers [73]. In immigrant Latino families, economic stressors combined with parental immigration-related stress have predicted increased behavioral difficulties in preschoolers [75].

Although studies in our review did not address specific sociocultural stressors, a small but growing body of literature shows that culturally relevant precursors of maternal depression in Latina mothers (e.g., stressful life events, discrimination, acculturative stress) have been associated with socioemotional maladjustment during early developmental windows, suggesting that maternal stress in general may affect a child's capacity for healthy socioemotional outcomes beginning as early as the prenatal period [38,76–78]. Moreover, bio-behavioral profiles indicative of chronic stress across succeeding generations have been found in Mexican-origin women, whereby third generation women exhibited blunted maternal cortisol during pregnancy [79].

These increasing socioemotional vulnerabilities echo our findings linking consistently poorer socioemotional outcomes to children of U.S.-born mothers compared to children of foreign-born mothers. Such outcomes occurred even within the context of higher family

incomes, maternal education, and higher levels of maternal sensitivity scores frequently found in succeeding generations [2]. A previous study examining school readiness also noted healthier socioemotional adjustment in kindergarten children of foreign-born Latino parents compared to children of U.S.-born parents of other races and ethnicities, as rated by teachers [80]. Similarly, a study examining the relationship between cultural adaptation in parents and behavioral functioning in preschoolers found that high US identity in Latino parents was associated with more externalizing behaviors, whereas higher ethnic parental identity predicted lower levels of child externalizing and internalizing behaviors [81]. Increased parental acculturation, as measured by language usage, has also been linked to lower (i.e., less adaptive) cortisol responses in preschoolers living in poverty [75]. Although cross-generational research supports an immigrant paradox in behavioral outcomes during adolescence (e.g., substance abuse, teen pregnancy, delinquency), research examining socioemotional outcomes in children under five years of age has been scarce [82]. This is the first review, to our knowledge, to have found support for the Latino paradox in early childhood mental health. If an immigrant advantage exists throughout early developmental windows in the socioemotional domain, why and when does it erode?

The interaction between maternal nativity and maternal mental health in relation to child outcomes was rarely examined in our studies, preventing us from fully grasping the underlying mechanisms for this apparent decline. However, our findings clearly indicate that a) this pattern can be observed within sensitive early developmental windows; b) maternal mental health and early child development are intimately connected; and c) contextual stressors are salient precursors of maternal mental health. Maternal biopsychosocial stress across generations stemming from the chronic stressors inherent in unequitable societal structures may be a salient contributing factor; protective cultural factors such as collectivism that confer psychological resilience [83] may also be lost in subsequent generations. Combining models based on risk (e.g., developmental-risk framework) and resilience (e.g., ecocultural theory) may help elucidate underlying mechanisms for the Latino Paradox [84].

## Clinical implications

Maternal depression is treatable, and its effects on child socioemotional outcomes can be minimized. A meta-analysis revealed that non-pharmaceutical treatment of depression during pregnancy not only improved maternal mood but improved child functioning–particularly infant regulatory capacity [85]. These outcomes provide causational support for the link between prenatal depression and infant socioemotional outcomes as well as subsequent evidence for the intergenerational transmission of mental health disparities. Furthermore, the World Health Organization recommends that interventions should treat maternal depression *and* support the mother-child relationship concurrently [24]. Well-designed, culturally responsive interventions that mitigate maternal stress and strengthen the mother-child relationship may also foster healthy child development. Although depression is treatable and support is critical during early developmental windows when interventions would be most preventative and cost-effective, treatment often remains inaccessible to Latina mothers even after controlling for income levels [6,86]. Given that maternal depression is highly correlated with adversity, lack of appropriate treatment for mothers suffering from depression and their young children may further compound disparities in children's socioemotional development. Policies should address the need to support maternal mental health as early as the prenatal period.

Effective interventions for maternal depression in Latinas have focused on culturally sensitive psychoeducation, emotional support, and access to community resources [87].

Exploratory family-level interventions have also embraced potential sociocultural strengths such as kinship and familism (i.e., family interdependence), resulting in decreased maternal depressive symptoms and decreased internalizing and externalizing behaviors in children [88]. Moreover, a culturally-tailored intervention targeting postpartum depression and mother-child interactions in infants and toddlers lowered maternal depression scores and improved maternal perceptions of the child's socioemotional competence [89]. Upstream approaches in immigrant women should address culturally relevant stressors and barriers that contribute to depression and stress, such as potential and poignant losses experienced during immigration: social support networks lost through family separation, a sense of belonging dislodged by discrimination, and access to community resources impeded by language barriers [41].

## Limitations

Due to scarce research examining maternal depression in Latinas and child socioemotional outcomes, we had a limited number of studies–five of which were cross-sectional. Studies were too heterogeneous to conduct a meta-analysis. Furthermore, most studies during the preschool years used maternal-reported measures of child behavior such as the CBCL. Behavior ratings may be influenced by maternal cultural expectations [20] and maternal depression. Mothers with depression may be more likely to view their children negatively, resulting in potentially biased ratings [1]. However, three out of five studies using maternal reported child measures in longitudinal studies adjusted for concurrent depression.

We also only included studies conducted in the US. Although this focus prevented confounding factors that may have arisen from research in several host countries, our findings cannot be generalized outside of the US. Moreover, the Latino population is not homogeneous; subpopulations from Latin American countries have various socioeconomic and cultural backgrounds as well as reasons for coming to the United States [90]. Furthermore, acculturation may differ among Latino sub-groups [38]. Although we extracted data such as country of origin and acculturation, eight studies did not include country of origin, and seven did not measure acculturation.

## Conclusions

This is the first systematic review to examine Latina mother-child dyads in relation to maternal depression and child socioemotional development. We found significant positive correlations between maternal depression and contextual stressors, inverse correlations between maternal depression and child well-being, and evidence of the moderating and mediating role of maternal depression between contextual stressors and child socioemotional outcomes. We also connected fragmented evidence across studies delineating a concerning and significant cross-generational decline in young Latino children's socioemotional outcomes; future research should examine the underlying mechanisms. Indeed, socioemotional developmental processes are critical throughout infancy and early childhood—processes allowing children to form friendships, learn in school, develop resiliency, and eventually strengthen our intrinsically interconnected society [91].

## Supporting information

**S1 Table. Prisma 2009 checklist.**
(PDF)

**S2 Table. Cognitive outcomes.**
(DOCX)

**S1 File. Protocol for systematic review.**
(PDF)

## Acknowledgments

We would like to thank Dr. Beth Black for editorial suggestions.

## Author Contributions

**Conceptualization:** Rebeca Alvarado Harris, Hudson P. Santos, Jr.

**Data curation:** Rebeca Alvarado Harris, Hudson P. Santos, Jr.

**Formal analysis:** Rebeca Alvarado Harris.

**Investigation:** Rebeca Alvarado Harris, Hudson P. Santos, Jr.

**Methodology:** Hudson P. Santos, Jr.

**Resources:** Hudson P. Santos, Jr.

**Supervision:** Hudson P. Santos, Jr.

**Writing – original draft:** Rebeca Alvarado Harris.

**Writing – review & editing:** Rebeca Alvarado Harris, Hudson P. Santos, Jr.

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
