## [Decision Letter · Decision Letter 0]

13 Nov 2019

PONE-D-19-24059

Maternal depression in Latinas and child socioemotional and cognitive development: A systematic review

PLOS ONE

Dear Dr. Santos Jr,

Thank you for submitting your manuscript to PLOS ONE. After careful consideration, we feel that it has merit but does not fully meet PLOS ONE’s publication criteria as it currently stands. Therefore, we invite you to submit a revised version of the manuscript that addresses the points raised during the review process.

We would appreciate receiving your revised manuscript by Dec 28 2019 11:59PM. To enhance the reproducibility of your results, we recommend that if applicable you deposit your laboratory protocols in protocols.io, where a protocol can be assigned its own identifier (DOI) such that it can be cited independently in the future. For instructions see: http://journals.plos.org/plosone/s/submission-guidelines#loc-laboratory-protocols

We look forward to receiving your revised manuscript.

Kind regards,

James Swain, M.D., Ph.D., F.R.C.P.C.

Academic Editor

PLOS ONE

Journal Requirements:

Additional Editor Comments (if provided):

Dear colleagues,

Your manuscript entitled "Maternal depression in Latinas and child socioemotional and cognitive development: A

systematic review" has now been reviewed and the reviewer comments are appended below. I agree with them that your work is of importance and interest - however, they have raised points that need to be addressed by a major revision.

Reviewers' comments:

Reviewer's Responses to Questions

**Comments to the Author**

1. Is the manuscript technically sound, and do the data support the conclusions?

Reviewer #1: Yes

Reviewer #2: Partly

2. Has the statistical analysis been performed appropriately and rigorously? 

Reviewer #1: I Don't Know

Reviewer #2: Yes

3. Have the authors made all data underlying the findings in their manuscript fully available?

Reviewer #1: Yes

Reviewer #2: Yes

4. Is the manuscript presented in an intelligible fashion and written in standard English?

Reviewer #1: Yes

Reviewer #2: No

5. Review Comments to the Author

Reviewer #1: This is a meta-analysis and systematic review of 18 studies that examined relationship between maternal depression in Latina mothers and their children’s social, emotional and cognitive outcomes over the first five years of life. The study advances the area of research on the effects of maternal depression on child development by addressing both economic and sociocultural contexts of family functioning. The main results of the study included inverse associations between maternal depression and child socioemotional and cognitive outcomes and that children of U.S.-born Latina mothers had poorer developmental outcomes than children of foreign-born Latina mothers. the study is also innovative in that this is the first systematic review of studies examining Latina mothers’ depression and child psychological outcomes. The authors may consider the following comments:

1) Given that socioemotional and cognitive outcomes reflect a very wide range of constructs, were there any inclusion criteria for the reliability of measures that were used to study these constructs in young children?

2) Maybe I did not see in this in the study inclusion criteria section of the paper, but was there a requirement for how depression was evaluated and diagnosed in mothers who participated in studies included in this review?

3) The quality assessment criteria in Table 1 follow the guidelines from the National Heart, Lung and Blood Institute. Are these criteria commonly used to determine the quality of studies assessing mental health outcomes?

4) Related to question 1 above, it is indicated on page 18 of the paper that most studies measured child’s socioemotional development by mother-rated scales? Is it possible that these ratings were affected by the mother’s depression?

5) The qualitative review of the literature if clearly described and Table 2 is very informative but more detail would help to understand the quantitative data analytic strategy. Is it possible to obtain the magnitude of effect sizes for the strength of association among the variables of interest?

Thank you for the opportunity to review this interesting and important paper.

Reviewer #2: Thank you for opportunity to review the manuscript "Maternal Depression in Latinas and child socioemotional and cognitive development: A systematic review.” The authors are addressing a very important topic with implications for psychological care for Latino populations for whom psychological interventions have failed to serve well historically. This manuscript has a number of primarily methodological strengths such as pre-registering the study, using the PRISMA guidelines, and coding to data quality which are all very important. However there are several concerns primarily in the introduction in terms of setting up the aims and scope of the review that detract from the manuscript’s potential.

General comments

1. Use consistent terms for your constructs (many different terms for the outcome throughout) and define your constructs upfront and all that they encompass for clarity. There are many examples of this but for instance you say you at just looking at 0-5 years of age but cite a handful with kids older than 5 and say you are only looking at maternal depression but cite a handful that combine stress and depression or depression and other psychopathology.

2. Mediation and moderation are distinct conceptually and have different implications so the many instances of grouping them together is misleading. Set them up separately and interpret them separately since they tell us different things.

Introduction

1. My main comment is that your outcome (essentially all biological social and psychological aspects of development) is far too broad to make a cohesive argument and defined differently in different places. You look at child neurodevelopment, language development, social development, emotion regulation and self regulation, psychophysiology (RSA, EEG, ANS reactivity), attachment and psychological problems (e.g. externalizing). It would be very hard to review the background literature on all of these things adequately in the introduction which makes for a premise that is not well set up. I would strongly recommend narrowing your focus and in particular getting rid of the one study here and there that looks at a different construct entirely than the others (e.g. attachment and psychophysiology). I would recommend focusing on psychological outcomes (social behavior and externalizing) and closely related constructs (e.g. emotion regulation) since the studies seem to overwhelmingly focus on this, its much easier to build a cohesive theory around this, and you can clearly define the literature and your construct for readers to begin with.

2. Related to defining your constructs clearly what is meant by economic hardship you go on to describe community violence, overcrowded housing, poverty (defined as below the poverty line I assume) as variables in the studies so make it clear these are all proxy for poverty that you will be considering under the umbrella of “economic hardship.”

3. Restructuring thoughts: Your paper would benefit from an integrated conceptual model presented early on (a figure would be helpful), then the evidence of the model in white families, then the evidence in LatinX families and the need for research in LatinX families. Instead you highlighted different pieces from different angles (eg. Maternal sensitivity paragraph, FSM paragraph, and acculturative stress paragraph). It seems that your full model is that acculturative stress and economic stress (poverty) leads to maternal depression which impacts poor child outcomes through compromised parenting. Then that we should focus on Latina moms because they disproportionately experience acculturative stress and poverty. You should also mention that not all of the studies cover all pieces of the model but they all have the maternal depression and child outcomes piece in Latina samples.

4. Make the argument or mothers vs. fathers. The argument is maternal depression has a robust impact on child outcomes prenatally an there are less studies on fathers I assume but this needs to be explicit.

5. You should comment on the immigrant paradox literature explicitly in the argument for Latinas specifically section.

6. Line 53: comment on depression rates in Latinx populations specifically

7. Developmental argument: I would recommend adding to your developmental windows section talking about why you chose 0-5 year olds specifically (e.g. include the age ranges in the studies you are citing) given that this is never said until the last sentence of the introduction. What is the evidence (not just the theory) that this range is integral? For example, you could cite here evidence that intervening on child outcomes for 0-5 year olds with depressed mothers improves child psychological outcomes long term. Also explain why you use 5 as a cutoff and not earlier? Attachment theory may also be helpful to augment this developmental argument. I love the developmental figure though, that was very helpful.

8. Relatedly don’t review literature on adolescent outcomes (line 112) if your focus is 0-5 years of age, this is inconsistent with your argument for focusing on 0-5.

9. Biology is not “permanent” as you say in line 70, there is overwhelming evidence of plasticity in biological responses which is hopeful for interventions.

Methods

1. Your methods were very strong on the whole and my main comment is taking out the less relevant articles as stated above since it detracts from being able to make tangible and focused conclusions but also have some minor comments below.

2. Line 157: did you exclude for psychological co-morbid diagnoses (e.g. anxiety etc)?

3. It may be helpful to describe the last date of searching databases so future researchers will know where to pick up for later reviews. For example “ we reviewed all studies before June of 2019.”

4. Line 233: brief description of the three bag and two bag paradigms would be helpful here.

5. You need to acknowledge the diversity of what could be meant by depression and continue to clarify what you mean by depression when you introduce a study. For instance, are we talking about an MDD diagnosis (or a PDD, or a past MDE) or depression symptoms on a continuum that may or may not reach clinical cutoffs (e.g. CESD).

6. Line 367: 71 months is more than 5 years so you should justify why it was included there are also there are other studies that include older children (such as in line 320).

7. Line 376: It doesn’t seem that the literature is mixed it seems that it is leaning towards there is a not a significant mediation by maternal depression.

8. Line 381: I don’t understand what the first part of the sentence is saying, what is the finding? I think there may be some words missing. Please clarify.

9. Give the acculturation findings their own cohesive paragraph instead of sprinkling them throughout multiple sections to improve clarity.

Discussion

1. Line 430: “contextual stressors in conjunction with depression heightened associations between maternal depression and child maladjustment” is not supported in what you found, you found 2 of these studies were not significant and one was but only when combined with stress and explaining the lack of findings would be helpful.

2. I would tie in the immigrant paradox literature (e.g. Garcia Coll’s work) again to interpret your finding that things get worse with greater acculturation (Lines 485-487). There is a lot of literature on this.

3. Interpret why depression alone did not predict child outcomes as strongly as depression and stress together. For instance this could go after line 468.

Minor points

1. Imprecise language should be specified for instance:

a. Line 53: should be prevalence rates of major depressive disorder (or whatever you define in your construct) for clarity.

b. Line 124 these findings are specific to first compared to second generation immigrants not “intergenerational” broadly.

c. Line 129 should be largest ethnic minority (it isn’t sexual or gender minorities or necessarily racial minority)

6. PLOS authors have the option to publish the peer review history of their article (what does this mean?). If published, this will include your full peer review and any attached files.

Reviewer #1: No

Reviewer #2: No

---

## [Author Response · Author response to Decision Letter 0]

9 Jan 2020

Please see our "Response to Reviewers" file.

---

## [Decision Letter · Decision Letter 1]

26 Feb 2020

Maternal depression in Latinas and child socioemotional development: A systematic review

PONE-D-19-24059R1

Dear Dr. Santos Jr,

We are pleased to inform you that your manuscript has been judged scientifically suitable for publication and will be formally accepted for publication once it complies with all outstanding technical requirements.

With kind regards,

James Swain, M.D., Ph.D., F.R.C.P.C.

Academic Editor

PLOS ONE

Additional Editor Comments (optional):

Reviewers' comments:

Reviewer's Responses to Questions

**Comments to the Author**

1. If the authors have adequately addressed your comments raised in a previous round of review and you feel that this manuscript is now acceptable for publication, you may indicate that here to bypass the “Comments to the Author” section, enter your conflict of interest statement in the “Confidential to Editor” section, and submit your "Accept" recommendation.

Reviewer #1: All comments have been addressed

Reviewer #2: All comments have been addressed

2. Is the manuscript technically sound, and do the data support the conclusions?

Reviewer #1: Yes

Reviewer #2: Yes

3. Has the statistical analysis been performed appropriately and rigorously? 

Reviewer #1: Yes

Reviewer #2: N/A

4. Have the authors made all data underlying the findings in their manuscript fully available?

Reviewer #1: Yes

Reviewer #2: Yes

5. Is the manuscript presented in an intelligible fashion and written in standard English?

Reviewer #1: (No Response)

Reviewer #2: Yes

6. Review Comments to the Author

Reviewer #1: All comments have been addressed and I think the paper will make a strong contribution to the area of effects of maternal depression on child development

Reviewer #2: The authors were very thorough in addressing my initial concerns and the manuscript as it stands is much clearer and more cohesive.

7. PLOS authors have the option to publish the peer review history of their article (what does this mean?). If published, this will include your full peer review and any attached files.

Reviewer #1: No

Reviewer #2: No

---

## [Editor Report · Acceptance letter]

2 Mar 2020

PONE-D-19-24059R1 

Maternal depression in Latinas and child socioemotional development: A systematic review 

Dear Dr. Santos Jr:

I am pleased to inform you that your manuscript has been deemed suitable for publication in PLOS ONE. Congratulations! Your manuscript is now with our production department. 

With kind regards,

on behalf of

Dr James Swain 

Academic Editor

PLOS ONE